# WavefrontDiffusion: Dynamic Decoding Schedule For Improved Reasoning

**Haojin Yang**[1], **Rui Zhou**[1], **Rui Hu**[2], **Zequn Sun**[2], **Yujun Cai**[3], **Yiwei Wang**[4]*

[1]School of Software and Microelectronics, Peking University
[2]State Key Laboratory for Novel Software Technology, Nanjing University
[3]The University of Queensland, [4]University of California, Merced
`yhj@stu.pku.edu.cn; yiweiwang2@ucmerced.edu`
https://github.com/page link

## Abstract

Diffusion Language Models (DLMs) have shown strong potential for text generation and are becoming a competitive alternative to autoregressive models. The denoising strategy plays an important role in determining the quality of their outputs. Mainstream denoising strategies include Standard Diffusion and BlockDiffusion. Standard Diffusion performs global denoising without restricting the update range, often finalizing incomplete context and causing premature end-of-sequence predictions. BlockDiffusion updates fixed-size blocks in a preset order, but its rigid structure can break apart coherent semantic units and disrupt reasoning. We present **WavefrontDiffusion**, a dynamic decoding approach that expands a wavefront of active tokens outward from finalized positions. This adaptive process follows the natural flow of semantic structure while keeping computational cost equal to block-based methods. Across four benchmarks in reasoning and code generation, WavefrontDiffusion achieves **state-of-the-art performance** while producing outputs with higher semantic fidelity, showing the value of adaptive scheduling for more coherent and efficient generation.

## 1 Introduction

Recent advances in large language models (LLMs) have achieved remarkable progress in complex reasoning and structured generation tasks such as mathematical problem solving and code synthesis (OpenAI et al., 2025; DeepSeek-AI et al., 2025). Autoregressive (AR) models remain the dominant paradigm for these tasks due to their stepwise logical consistency (Deletang et al., 2024). However, their strictly sequential nature introduces latency and limits flexibility, which can be problematic in settings that demand both accuracy and responsiveness, such as interactive assistants or real-time code generation. These limitations have motivated the exploration of alternative decoding paradigms that can balance quality, efficiency, and adaptability (Leviathan et al., 2023).

Diffusion Language Models (DLMs) have recently emerged as a promising alternative by framing text generation as an iterative denoising process (Gong et al., 2025; Song et al., 2025). Unlike AR models, which finalize one token per step, DLMs update multiple tokens in parallel over a series of iterations (Schiff et al., 2025; Lou et al., 2024). This design enables more flexible generation while maintaining global coherence, offering potential gains in both efficiency and representational capacity(Sahoo et al., 2024).

The capabilities of DLMs partly rely on the design of their denoising schedule, yet this aspect has received limited exploration. Current mainstream strategies include *Standard Diffusion*(Chang et al., 2022) and *BlockDiffusion*(Arriola et al., 2025). Standard Diffusion applies a global and uniform denoising policy without restricting the update range, which often causes overconfidence in end-of-sequence tokens(Nie et al., 2025) and insufficient context for denoised tokens, leading to early errors that cannot be corrected. Block Diffusion improves stability by processing fixed-size blocks in a preset order, but it does not account for the natural semantic boundaries of the output, causing the denoising of coherent sequences to be artificially fragmented.

---

*Corresponding author.

Figure 1: Illustration of scheduling strategies in diffusion language models. A Diffusion Language Model predicts all masked tokens. At each step, a scheduling strategy decides which tokens to denoise. There are three strategies: Standard Diffusion uses global scheduling. It always selects tokens from all masked positions. BlockDiffusion limits the choice to a candidate scope. The scope is fixed blocks, similar to semi-autoregression. **WavefrontDiffusion** also uses a candidate scope. Unlike BlockDiffusion, the scope grows dynamically during decoding.

To address these limitations, a new denoising strategy should offer several key benefits. First, it should support **adaptive scheduling**, dynamically adjusting the denoising process to follow the evolving generation context rather than relying on fixed update patterns. Second, it should provide each token with more **complete contextual information** during denoising, enabling smoother and more coherent outputs. Finally, it should **preserve computational efficiency**, matching the cost of block-based methods while delivering higher quality generation.

We introduce **WavefrontDiffusion**, a dynamic scheduling strategy for DLMs designed to overcome the limitations of prior approaches. It adaptively expands a *wavefront* of active positions, allowing the denoising process to follow the natural structure of the generation context. Each token is updated with richer and more complete contextual information, producing smoother and more coherent outputs. At the same time, WavefrontDiffusion maintains the computational efficiency of block-based methods, delivering higher quality generation without additional cost.

**Contributions.** Our contributions are fourfold:

1. We identify the key limitations of existing denoising schedules, including insufficient context in Standard Diffusion and the rigid boundaries of Block Diffusion that fragment coherent sequences.

2. We propose **WavefrontDiffusion**, a training-free dynamic scheduling method that adapts to the generation context without introducing any additional computational cost.

3. We conduct comprehensive experiments across four reasoning and code generation benchmarks, demonstrating clear improvements in output correctness and generation quality compared to strong baselines.

4. We perform a detailed hyperparameter analysis to validate the reliability and robustness of WavefrontDiffusion under different settings.

## 2 BACKGROUND

This section provides background on Diffusion Language Models (DLMs) and two decoding schedules: Standard Diffusion and BlockDiffusion. These methods serve as baselines for comparison with our proposed WavefrontDiffusion.

### 2.1 PRELIMINARIES: DIFFUSION LANGUAGE MODELS

Diffusion Language Models (DLMs) treat text generation as an iterative denoising process over a sequence of discrete tokens (Austin et al., 2021a). Let $\mathbf{x}_0 = (x_{0,1}, \ldots, x_{0,N})$ denote a clean token

sequence of length $N$. The forward *corruption process* gradually replaces tokens with a special mask token `[MASK]`, following a masking probability $t \in [0, 1]$:

$$q(\mathbf{x}_t \mid \mathbf{x}_0) = \prod_{i=1}^{N} \left[ (1-t)\,\delta(x_{t,i} = x_{0,i}) + t\,\delta(x_{t,i} = \texttt{[MASK]}) \right],$$

where $\delta(\cdot)$ is the Dirac delta function. Intuitively, each token is independently masked with probability $t$. The reverse process is parameterized by a model $p_\theta$, which learns to iteratively *denoise* the corrupted sequence by predicting less-noised states:

$$p_\theta(\mathbf{x}_{t-1} \mid \mathbf{x}_t, \mathbf{c}) = \prod_{i=1}^{N} p_\theta(x_{t-1,i} \mid \mathbf{x}_t, \mathbf{c}),$$

where $\mathbf{c}$ represents the conditioning context, such as a prompt or input sequence. Training minimizes the cross-entropy loss over masked positions under a variational lower-bound (VLB) objective. At inference time, generation starts from a fully masked sequence $\mathbf{x}_1 = \texttt{[MASK]}^N$ and iteratively applies $p_\theta$ until a clean sequence is produced.

## 2.2 STANDARD DIFFUSION DECODING

**Standard Diffusion** follows a mask-prediction process similar to *MaskGIT*. At each step, the model predicts tokens for all masked positions in parallel:

$$\mathbf{p}_t = p_\theta(\mathbf{x}_{\text{masked}} \mid \mathbf{x}_t, \mathbf{c}),$$

where $\mathbf{c}$ is the conditioning context and $\mathbf{x}_t$ is the current partially generated sequence. From these predictions, only a subset of tokens with the highest confidence is selected and finalized, while the remaining positions stay masked for future updates.

This process repeats until every position is finalized. Standard Diffusion allows flexible, non-sequential generation without fixed update order. However, because selection relies solely on local confidence, early errors may be locked in too soon and propagate through later steps. This often results in premature end-of-sequence (EOS) tokens and logically inconsistent outputs, especially in reasoning tasks that require multi-step coherence.

Standard Diffusion predicts all masked positions in parallel but updates only the most confident ones at each step. It is simple and efficient but prone to cascading errors due to the lack of global structural guidance.

## 2.3 BLOCK-WISE DIFFUSION SCHEDULING

**BlockDiffusion** limits the search space for each denoising step by dividing the sequence into $K$ contiguous, non-overlapping blocks $\{\mathcal{B}_1, \ldots, \mathcal{B}_K\}$. The model generates tokens block by block in a fixed left-to-right order, where $\mathbf{p}$ is the initial prompt context.:

$$p(\mathcal{B}_k \mid \mathbf{p}, \mathcal{B}_1, \ldots, \mathcal{B}_{k-1}),$$

Only tokens inside the current block are considered for updating at each step. This restriction helps control error propagation and provides a more stable decoding process than Standard Diffusion. However, the fixed block boundaries and update order introduce two key problems: **Semantic fragmentation:** Block boundaries can split coherent structures such as function signatures, equations, or reasoning steps, which forces the model to finalize incomplete semantic units. **Lack of adaptivity:** The update order cannot change according to context or token-level confidence, which reduces flexibility during generation.

BlockDiffusion improves stability but remains limited for complex reasoning tasks. Semantic units often vary in length and dependencies can span multiple blocks, making a fixed schedule suboptimal. This limitation motivates the adaptive scheduling strategy we introduce in Section 3.

# 3 WAVEFRONTDIFFUSION METHOD

This section introduces **WavefrontDiffusion**, a new adaptive scheduling strategy for discrete diffusion language models. Unlike fixed-block decoding, which relies on pre-defined partitions, WavefrontDiffusion dynamically expands a localized frontier of candidate positions. This approach better aligns with the semantic structure of the output while maintaining identical computational cost.

## 3.1 METHOD THEORY

The decoding process of diffusion language models (DLMs) relies heavily on the *scheduling policy*, which determines the order in which masked tokens are finalized. A well-designed schedule should align with the natural semantic flow of generation while maintaining computational efficiency. However, existing methods fall short in this regard. Standard Diffusion updates all positions simultaneously, often finalizing incomplete or incorrect context. BlockDiffusion improves stability by partitioning the sequence into fixed-size blocks, but its rigid structure *artificially fragments* semantic units such as reasoning steps or code segments. This misalignment between block boundaries and true semantic boundaries leads to premature commitments and cascading errors.

**Intuition: generation as a spreading wave.** To address these limitations, we introduce the concept of a *wavefront*—a dynamic set of candidate positions that expands outward from already finalized tokens. Instead of processing fixed contiguous blocks, the decoding process grows like a wave, gradually extending into surrounding masked regions as local context becomes available. This design ensures that each token is updated only when it has access to sufficient contextual information, thereby respecting semantic locality and preventing incomplete updates. Unlike static schedules, a wavefront adapts to the evolving structure of the sequence, allowing the model to prioritize semantically relevant regions at each step.

**Theoretical Grounding.** We formalize this intuition through the **Information Gradient Hypothesis**, which posits that the conditional entropy of a token increases monotonically with its distance from the finalized context. While BlockDiffusion restricts the search space to rigid pre-defined blocks, WavefrontDiffusion performs an optimal restricted search within "entropy isosurfaces" defined by this distance. We prove in **Appendix D** that our dynamic boundary is mathematically guaranteed to contain a higher density of low-entropy candidates compared to static blocks, thereby minimizing the probability of semantic mismatch.

**Mathematical definition.** Let $N$ be the total length of the output sequence and $\mathcal{C}_t$ denote the set of finalized tokens at iteration $t$. We define the *wavefront set* $\mathcal{W}_t$ as:

$$\mathcal{W}_t = \{i \mid \mathrm{dist}(i, \mathcal{C}_t) \leq R\}, \tag{1}$$

where $\mathrm{dist}(i, \mathcal{C}_t)$ measures the minimum distance between position $i$ and any finalized position, and $R$ is a user-defined *expansion radius*. Intuitively, the wavefront contains all masked tokens that are within $R$ steps of a completed token, ensuring that updates focus on tokens with nearly complete local context.

At the beginning of decoding, $\mathcal{C}_0$ contains only the prompt context, and $\mathcal{W}_0$ is initialized as the first $F$ positions following the prompt, where $F$ is the maximum wavefront size. As generation proceeds, $\mathcal{W}_t$ expands dynamically based on the evolving structure of $\mathcal{C}_t$. This expansion process adaptively shifts computational focus to regions that are semantically important, while maintaining strict limits on total computational cost.

This dynamic frontier design provides two key benefits:

1. **Context completeness:** Tokens are only finalized when surrounded by sufficient context, mitigating the boundary truncation problem in BlockDiffusion.

2. **Compute parity:** By capping the wavefront size at $F$, the overall number of token updates remains identical to block-based methods, ensuring no increase in computational cost.

In the next subsection, we will provide a step-by-step algorithmic description of how WavefrontDiffusion operationalizes this theory into a practical decoding strategy.

## 3.2 Method: Algorithmic Details

Building on the theoretical foundation in Section 3.1, we now describe the concrete implementation of WavefrontDiffusion. The goal is to operationalize the dynamic wavefront mechanism into a practical, step-by-step decoding strategy. At a high level, WavefrontDiffusion iteratively updates a subset of masked positions, allowing the denoising frontier to expand outward from previously finalized tokens while strictly controlling computational cost. This ensures that tokens are finalized in a context-aware manner without increasing the total number of forward passes compared to block-based schedules.

**Core iterative process.** Each iteration of WavefrontDiffusion consists of four sequential steps, summarized below and illustrated in Algorithm 1.

1. **Score.** Perform a single forward pass of the model to compute logits for all masked positions. For each masked position $j$, compute a confidence score $s_j$ as the maximum softmax probability:

$$s_j = \max_{v \in \mathcal{V}} p_\theta(x_j = v \mid x_t, c), \tag{2}$$

   where $\mathcal{V}$ is the vocabulary and $c$ is the conditioning context (e.g., prompt). These scores are cached for later use.

2. **Select & Denoise.** From the current wavefront $\mathcal{W}_{t-1}$, select the top-$k_t$ positions with the highest confidence scores and finalize them by replacing their mask tokens with the predicted values:

$$k_t = k_{\text{base}} + \mathbb{1}[t \leq \text{extra}], \tag{3}$$

   where $k_{\text{base}} = \lfloor N/T \rfloor$ evenly divides the total $N$ tokens over $T$ steps, and `extra` distributes the remaining tokens. If $|\mathcal{W}_{t-1}| < k_t$, additional high-confidence positions outside the frontier are included to meet the per-step budget.

3. **Expand.** For each finalized position $i$, add all masked neighbors within a radius $R$ to the next wavefront $\mathcal{W}_t$. This adaptively expands the active frontier around newly completed regions:

$$\mathcal{W}_t = \bigcup_{i \in \mathcal{C}_t} \{j \mid \text{dist}(j, i) \leq R, \ x_j = [\texttt{MASK}]\}. \tag{4}$$

4. **Prune.** If $|\mathcal{W}_t| > F$, retain only the top-$F$ positions by confidence score. This strictly bounds the per-step computational footprint, ensuring parity with block-based methods.

This four-step cycle repeats until all tokens are finalized or the step budget $T$ is exhausted.

The full WavefrontDiffusion procedure is formalized in Algorithm 1. Here, $F$ denotes the maximum wavefront size, and $R$ controls the expansion distance of the frontier.

---

**Algorithm 1** WavefrontDiffusion Decoding

---

1: **Input:** model $\theta$, prompt context **c**, generation length $N$, total steps $T$, max wave size $F$, expansion radius $R$
2: Initialize masked sequence of length $N$ and wavefront $\mathcal{W}_0$ with the first $F$ positions
3: $k_{\text{base}} = \lfloor N/T \rfloor$, $\quad extra = N \bmod T$
4: **for** $t = 1$ to $T$ **do**
5: $\quad k_t = k_{\text{base}} + \mathbf{1}[t \leq extra]$ $\qquad\qquad\qquad\qquad\qquad\qquad$ ▷ Per-step budget
6: $\quad$ Run one forward pass to compute logits for all masked positions
7: $\quad$ Compute confidence scores $s_j$ for each masked position
8: $\quad$ Finalize these positions by replacing masks with $\arg\max$ predictions
9: $\quad$ Expand $\mathcal{W}_t$ with neighbors within radius $R$ of finalized positions
10: $\quad$ Prune $\mathcal{W}_t$ to size $F$ using cached confidence scores
11: **end for**

---

The total number of token updates performed by WavefrontDiffusion is strictly bounded by the product $F \times T$, which matches the compute budget of BlockDiffusion. The only difference lies in the *location* of updates, not their quantity. This design guarantees that improvements in quality arise solely from better scheduling rather than increased computational resources.

WavefrontDiffusion achieves dynamic, context-aware generation by repeatedly scoring, selecting, expanding, and pruning candidate tokens. By strictly limiting the wavefront size and per-step budget, the method maintains identical computational cost to block-based baselines while producing semantically coherent outputs.

## 4 EXPERIMENTS

We conduct a series of experiments to rigorously evaluate the effectiveness of WavefrontDiffusion and to answer the following key research questions:

1. **RQ1: Performance.** Does WavefrontDiffusion achieve higher accuracy on challenging reasoning and code generation benchmarks compared to compute-matched baselines?

2. **RQ2: Semantic Fidelity.** Does WavefrontDiffusion produce outputs that are more semantically coherent and faithful to ground truth, as measured by semantic similarity metrics?

3. **RQ3: Qualitative Behavior.** Can we qualitatively demonstrate that WavefrontDiffusion better respects semantic boundaries during generation, avoiding the forced segmentation observed in rigid block scheduling?

The following sections are organized to address these questions in order: Section 4.2 presents benchmark performance results (RQ1), Section 4.3 evaluates semantic fidelity using BERTScore (RQ2), and Section 4.4 presents a quantitative analysis of MHCO to demonstrate how WavefrontDiffusion better respects semantic boundaries during generation (RQ3).

### 4.1 EXPERIMENTAL SETUP

**Tasks and Datasets.** We evaluate WavefrontDiffusion on five benchmarks that test both mathematical reasoning and program synthesis: GSM8K(Cobbe et al., 2021), MATH(Saxton et al., 2019), HumanEval(Chen et al., 2021), MBPP(Austin et al., 2021b), and Big Bench Hard (BBH)(Suzgun et al., 2022). These datasets cover diverse domains and represent challenging evaluation scenarios for diffusion-based decoding.

All experiments are performed in a strict *zero-shot* setting without chain-of-thought prompting or few-shot examples. Any intermediate reasoning traces are generated by the model itself rather than injected hints. For reasoning benchmarks (GSM8K, MATH, BBH) we report accuracy using *exact match* (EM), and for HumanEval and MBPP we report *pass@1* accuracy. For semantic quality evaluation (Section 4.3), we use the WikiText dataset. We randomly sample 1,000 sentences and repeat evaluations several times, then report the averaged scores.

**Model and Baselines.** We use three discrete diffusion language models: **LLaDA-8B-Instruct**, **LLaDA-1.5**, and **Dream-7B** (Ye et al., 2025). Keeping the same backbone across all methods ensures that performance differences come only from the decoding schedule. We compare WavefrontDiffusion with two baselines: 1) **Standard Diffusion**, which updates all masked tokens in parallel at every step. This highlights the effect of structured scheduling. 2) **BlockDiffusion**, which denoises fixed, non-overlapping blocks in a rigid order and is the current state-of-the-art among block-based schedules.

**Evaluation Protocol.** All experiments run in a *deterministic environment* with temperature set to 0.0. This setting removes randomness and guarantees reproducibility, so we do not report standard deviations or seed-based variation in the main tables. The computational budget is controlled by fixing the number of forward steps to **1024** for every method. We measure cost as the *wall-clock runtime* needed to complete evaluation on each dataset. Because runtime can vary slightly across runs, the reported cost may differ by a small margin even when the number of forward steps is identical. All core hyperparameters (wavefront size $F$, expansion radius $R$, and step budget $T$) are provided in the Appendix to ensure transparency and reproducibility.

### 4.2 MAIN RESULTS

We first evaluate whether WavefrontDiffusion improves reasoning and code generation accuracy when the compute budget is fixed. Table 1 shows results across five benchmarks. WavefrontDif-

Table 1: **Main Performance Comparison.** Each dataset reports accuracy (Acc) and computation cost time (Cost). Highlighted rows indicate our proposed **WavefrontDiffusion**, which consistently outperforms Block-Diffusion under identical compute budgets. MBPP and Dream-7B results are added to demonstrate generalization.

| Strategy | GSM8K | | MATH | | HumanEval | | MBPP | | BBH | |
|---|---|---|---|---|---|---|---|---|---|---|
| | Acc | Cost | Acc | Cost | Acc | Cost | Acc | Cost | Acc | Cost |
| **LLaDA-8B-Instruct** | | | | | | | | | | |
| Standard Diffusion | 23.15 | 110.1 | 26.60 | 108.8 | 17.68 | 115.8 | 13.50 | 109.1 | 11.30 | 122.7 |
| BlockDiffusion | 80.74 | 113.6 | 40.62 | 109.4 | 45.73 | 107.2 | 41.17 | 109.2 | 43.23 | 116.6 |
| WavefrontDiffusion (Ours) | **82.03** | 112.2 | **41.04** | 109.9 | **47.56** | 109.3 | **42.40** | 107.8 | **44.30** | 115.1 |
| **LLaDA-1.5** | | | | | | | | | | |
| Standard Diffusion | 30.17 | 111.9 | 27.04 | 111.4 | 34.76 | 118.0 | 17.04 | 114.4 | 17.12 | 127.2 |
| BlockDiffusion | 82.33 | 111.6 | 41.64 | 107.4 | 46.34 | 110.1 | 44.04 | 118.1 | 44.56 | 118.6 |
| WavefrontDiffusion (Ours) | **82.94** | 109.3 | **41.96** | 110.9 | **48.17** | 112.9 | **46.20** | 112.3 | **45.26** | 114.1 |
| **Dream-7B** | | | | | | | | | | |
| Standard Diffusion | 35.03 | 99.8 | 29.98 | 100.8 | 20.12 | 99.5 | 25.05 | 99.1 | 16.27 | 103.2 |
| BlockDiffusion | 78.92 | 99.2 | 43.60 | 103.5 | 53.05 | 100.6 | 58.52 | 95.9 | 45.13 | 100.5 |
| WavefrontDiffusion (Ours) | **80.66** | 101.7 | **44.00** | 98.0 | **54.27** | 101.4 | **59.03** | 95.7 | **46.91** | 105.7 |

fusion achieves the best performance on all tasks and for all three model families. It outperforms Standard Diffusion by a large margin and also provides consistent gains over BlockDiffusion.

**Analysis.** For the larger **LLaDA-8B-Instruct**, WavefrontDiffusion improves accuracy by **+1.27** on GSM8K, **+0.42** on MATH, **+1.83** on HumanEval, **+1.23** on MBPP, and **+1.07** on BBH compared to BlockDiffusion. For the smaller **LLaDA-1.5**, the improvements are **+0.61** on GSM8K, **+0.32** on MATH, **+1.83** on HumanEval, **+2.16** on MBPP, and **+0.70** on BBH. We observe similar consistent gains on the **Dream-7B** model (e.g., **+1.74** on GSM8K, **+1.22** on HumanEval, and **+0.51** on MBPP), confirming that the benefits of our method generalize across different architectures.

These gains are obtained with the same number of forward steps (**1024**) and under the same runtime budget. Although the reported wall-clock cost may differ slightly due to measurement noise, the computational load is identical across methods.

The results highlight two key points. First, adaptive scheduling helps the model finalize tokens only when sufficient context is available. This reduces premature commitments that often occur in BlockDiffusion and improves answer correctness. Second, the improvements appear consistently across both mathematical reasoning and program synthesis, and across different model sizes and families, showing that the benefits of dynamic scheduling are general and not tied to scale. Together, these findings confirm that the performance gains come from better scheduling rather than additional compute.

## 4.3 SEMANTIC QUALITY EVALUATION

To address RQ2, we test whether WavefrontDiffusion produces outputs that are more faithful and coherent than BlockDiffusion. We use **BERTScore** (Zhang et al., 2020), which compares system outputs with human references through contextual embeddings. Recall (R) reflects completeness of the output, Precision (P) reflects avoidance of irrelevant tokens, and F1 balances both.

**Analysis.** WavefrontDiffusion reaches the highest scores across all metrics. The gain in Precision shows that it avoids inserting irrelevant or noisy tokens, while the gain in Recall shows that it completes sequences more fully. Together these improvements lead to a stronger F1. These results confirm that the wavefront strategy helps the model wait until enough context is available before finalizing tokens. This behavior preserves semantic boundaries and reduces the fragmentation that is common in block-based schedules. The findings give direct evidence that adaptive scheduling improves not only task accuracy but also the quality and coherence of generated text.

Table 2: BERTScore results on WikiText. Higher is better. WavefrontDiffusion achieves consistent gains in Precision, Recall, and F1.

| Strategy | F1 | P | R |
|---|---|---|---|
| Standard Diffusion | $0.7885 \pm 0.0045$ | $0.7664 \pm 0.0064$ | $0.7913 \pm 0.0018$ |
| BlockDiffusion | $0.7946 \pm 0.0035$ | $0.7663 \pm 0.0055$ | $0.8142 \pm 0.0020$ |
| WavefrontDiffusion (Ours) | $\mathbf{0.8094} \pm 0.0033$ | $\mathbf{0.7749} \pm 0.0052$ | $\mathbf{0.8236} \pm 0.0018$ |

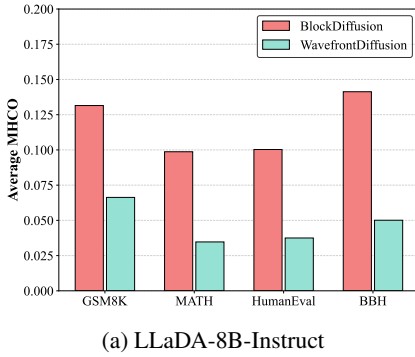

(a) LLaDA-8B-Instruct

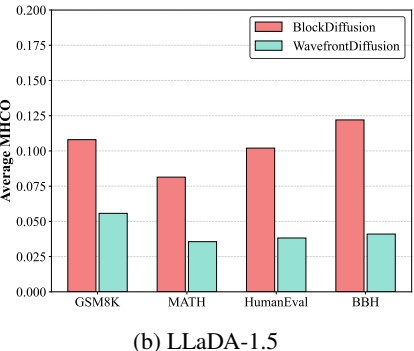

(b) LLaDA-1.5

Figure 2: Comparison of MHCO values across strategies. Lower values mean fewer priority violations and closer alignment with semantic order.

## 4.4 MEASURING PRIORITY VIOLATIONS VIA MHCO

To address RQ3, we introduce the **Masked Higher-Confidence Outside (MHCO)** metric. This metric measures how often the decoding process finalizes a low-confidence token before a nearby token with higher confidence. A lower MHCO score means that the schedule respects confidence ordering and is more consistent with semantic boundaries.

**Definition.** Let $S_t$ be the set of tokens selected for denoising at step $t$. Let $\mathcal{N}_{\text{out}}$ be the set of masked tokens within the expansion radius $R$ of the active frontier. We define:

$$\text{MHCO}_t = \frac{1}{|S_t|} \sum_{i \in S_t} \mathbf{1} \Big[ \exists j \in \mathcal{N}_{\text{out}} : c_t(j) > c_t(i) \Big],$$

where $c_t(i)$ is the confidence assigned to token $i$. A smaller value indicates that higher-confidence tokens are prioritized more consistently.

**Results.** Figure 2 shows that WavefrontDiffusion produces consistently lower MHCO values than BlockDiffusion on all datasets and both model scales. This means that the dynamic wavefront more often respects local confidence ordering.

These results give quantitative support for the intuition behind WavefrontDiffusion. The method finalizes tokens only when they are well supported by context and by confidence. This prevents the rigid boundary effects seen in BlockDiffusion, where tokens inside a block may be forced ahead of higher-confidence tokens just outside. Lower MHCO correlates with the accuracy gains in Table 1, showing that reducing priority mismatches leads to more coherent reasoning and better answers.

## 4.5 HYPERPARAMETER ANALYSIS

WavefrontDiffusion depends on two main hyperparameters: the maximum wavefront size $F$ and the expansion radius $R$. The parameter $F$ controls how many candidate tokens are considered in each step. The parameter $R$ controls how far the frontier can expand beyond finalized tokens. We study their impact while keeping all other settings fixed.

**Effect of $F$.** We fix $R = 2$ and test $F \in \{4, 8, 16\}$. Table 3 shows that accuracy increases when $F$ grows from 4 to 8. A larger frontier allows the model to consider more candidate positions and

finalize tokens that are better supported by context. However, accuracy does not improve further at $F = 16$. This indicates diminishing returns and suggests that very large frontiers may add redundant candidates without adding useful information.

**Effect of $R$.** We fix $F = 8$ and test $R \in \{2, 4, 8\}$. As Table 3 shows, expanding the radius from 2 to 4 yields small gains by letting the frontier reach semantically relevant positions earlier. Increasing $R$ further to 8 does not bring consistent improvements and can reduce accuracy on HumanEval. This suggests that an overly wide frontier may weaken local focus and introduce noise.

The results show that WavefrontDiffusion is stable across a wide range of values. Moderate settings such as $F = 8$ and $R = 2$ provide a good balance between local precision and global coverage. These settings also match our default configuration. Because the method is not highly sensitive to either parameter, it requires little tuning and can be deployed without major adjustments.

Table 3: Performance of WavefrontDiffusion under different hyperparameter settings. Accuracy remains stable across a wide range of $R$ and $F$, showing that the method is robust.

<div>

(a) Varying $F$ with $R = 2$

| Dataset | $F = 4$ | $F = 8$ | $F = 16$ |
|---|---|---|---|
| MATH | 41.02 | 41.04 | 41.22 |
| GSM8K | 82.71 | 82.03 | 82.03 |
| HumanEval | 45.12 | 47.56 | 45.12 |

(b) Varying $R$ with $F = 8$

| Dataset | $R = 2$ | $R = 4$ | $R = 8$ |
|---|---|---|---|
| MATH | 41.04 | 40.98 | 41.00 |
| GSM8K | 82.03 | 82.03 | 82.09 |
| HumanEval | 47.56 | 46.34 | 42.07 |

</div>

**Discussion.** In practice, we recommend using a conservative $F$ (e.g., $F = 8$) and adjusting $R$ based on task complexity and compute resources. This keeps tuning overhead low and ensures reliable performance. Despite these advantages, WavefrontDiffusion shares limitations with other diffusion-based decoders. It depends on internal confidence scores, which can be miscalibrated, especially out of domain. It also cannot fully avoid error propagation when early mistakes occur in long reasoning chains. Future work may explore delayed-finalization or reversible decoding to reduce these risks, and improved calibration methods to enhance robustness. Extending the approach to multi-modal or structured domains, such as code or graphs, is another promising direction.

## 5 RELATED WORK

This work relates closely to several areas: Diffusion Language Models (DLMs), decoding strategies for text generation, and confidence-based generation methods.

**Diffusion Language Models** Diffusion models originated in image generation (Ho et al., 2020; Song & Ermon, 2020). Recent research extends these principles to text, leading to Diffusion Language Models (DLMs) (Li et al., 2022). DLMs generate text through an iterative denoising process, showing promise in various text generation tasks (Ye et al., 2025; Xie et al., 2025). Early DLMs employed global masking strategies, predicting all masked tokens in parallel at each denoising step(Li et al., 2023). DLMs offer advantages in long text generation and mitigating exposure bias (Zeng et al., 2025).

**Decoding Strategies for DLM** Decoding strategies in text generation optimize output quality. Autoregressive models commonly use greedy decoding, beam search, or sampling methods (Meister et al., 2022). These methods typically generate tokens sequentially. For non-autoregressive models, particularly DLMs, decoding strategies are crucial. Existing DLM decoding strategies include: **Standard Diffusion**: This method repredicts all masked tokens at each timestep, lacking exploitation of already finalized tokens. **Block Diffusion**: This strategy partitions the sequence into fixed blocks for parallel generation. However, it can disrupt semantic coherence due to rigid boundaries. These strategies often involve trade-offs between generation quality and computational efficiency.

**Confidence-Based Generation** Confidence-based generation methods leverage a model's certainty in its predictions to guide the generation process. Such approaches are common in non-autoregressive models, enhancing both generation efficiency and quality (Gulrajani & Hashimoto, 2023; Li et al., 2025). For instance, some methods select high-confidence tokens for finalization

based on prediction certainty, focusing subsequent steps on lower-confidence regions. This strategy prioritizes stabilizing accurate predictions, thereby reducing error accumulation.

## 6 CONCLUSION

We introduced **WavefrontDiffusion**, a dynamic scheduling method for diffusion language models. The method balances semantic coherence and local accuracy in an adaptive way. Unlike static schedules, it uses token-level confidence to guide the expansion process. This design allows more effective exploration of the output space.

Experiments on mathematical reasoning and code generation benchmarks show that WavefrontDiffusion improves accuracy while preserving the computational cost of block-based decoding. We also proposed the *Multi-Hop Coherence* (MHCO) metric. This metric evaluates semantic consistency across generations and offers insights that standard accuracy cannot capture.

WavefrontDiffusion improves semantic coherence but depends on reliable token-level confidence. Poor calibration can reduce performance. Current experiments focus on zero-shot settings. Extending the method to few-shot or chain-of-thought prompting is an open direction. The computational cost remains similar to block decoding, but very long contexts may cause efficiency issues.

Future work includes three directions. First, improve confidence estimation to increase the reliability of dynamic scheduling. Second, explore alternative guidance signals for the denoising path, such as entropy or other uncertainty-based measures. Third, apply the method to domains such as program synthesis and long-form text generation. We expect WavefrontDiffusion to encourage broader research on adaptive decoding for diffusion-based models.

## ACKNOWLEDGEMENTS

We thank the anonymous reviewers for their constructive comments and suggestions, which helped improve the clarity and quality of this work. The work is partially supported by the U.S. National Science Foundation (NSF) Grant CRII 2451683, an NVIDIA Academic Grants Program, a U.S. Bank Academic Research Award, the University of California, Merced, and a UC Merced Faculty Research Award. The views and conclusions are those of the authors and do not necessarily reflect the official policy or position of the U.S. Government.

## ETHICAL CONSIDERATIONS

This research employs large language models for text generation. While our method primarily focuses on improving the technical performance of diffusion language models for tasks like reasoning and code generation, it is crucial to acknowledge the broader ethical implications associated with such technologies.

Our work, like all research involving generative AI, may contribute to the development of models that could potentially be used for generating misinformation, harmful content, or facilitating automated unethical activities. This includes, but is not limited to, the creation of deceptive text, biased narratives, or malicious code. We emphasize that the responsible development and deployment of these models require careful consideration of societal impact.

We adhere to the principles of responsible AI research. Our objective is to advance the capabilities of language models in beneficial domains, such as assisting in complex problem-solving and code development. We strongly advocate for the use of our findings for positive and ethical applications, and we urge future researchers and practitioners to critically assess and mitigate potential harms arising from the misuse of generative AI technologies.

Furthermore, any generative model may inadvertently perpetuate biases present in its training data. While our method is a decoding strategy and does not introduce new biases into the base model itself, it is dependent on the underlying model's inherent characteristics. Future work should continue to explore methods for detecting and mitigating such biases in the complete generation pipeline.

## REPRODUCIBILITY STATEMENT

We confirm that all experiments reported in this paper can be reproduced using the provided code and environment. The exact training and evaluation procedures, including dataset preprocessing, hyperparameter settings, and evaluation scripts, are included in our submission. We will release the full codebase and necessary instructions to the public after the review process to ensure transparency and facilitate future research.

## LANGUAGE MODEL USAGE STATEMENT

This paper used a large language model (LLM) solely as a tool to assist in code refinement and writing polishing. All core ideas, experiments, results, and analyses originate from the authors. The authors fully reviewed and validated every change introduced by the LLM to ensure correctness and avoid hallucinations.

No section of the research, including theoretical contributions or experimental design, was authored or invented by the LLM. Responsibility for all content rests entirely with the authors, who confirm that no misleading claims, fabricated data, or misrepresentations are present due to the use of the LLM.

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

APPENDIX

# A  HYPERPARAMETER INTRODUCTION

This section provides a comprehensive overview of all hyperparameters utilized in the Wavefront Diffusion algorithm and its experimental setup.

## A.1  WAVEFRONT DIFFUSION SPECIFIC PARAMETERS

| Parameter | Description |
|---|---|
| $F$ | The maximum size of the wavefront. This integer parameter defines the maximum number of masked tokens within the wavefront that can be simultaneously considered for denoising. A larger $F$ allows for broader contextual consideration but may increase computational overhead per step. |
| $R$ | The expansion radius. This integer parameter determines how many tokens outwards from a newly denoised (finalized) token the wavefront can expand. It controls the locality of the wavefront's growth. A larger $R$ facilitates faster wavefront propagation. |

## A.2  GENERAL DIFFUSION LANGUAGE MODEL PARAMETERS

| Parameter | Description |
|---|---|
| $T$ | The total number of diffusion timesteps. This integer parameter defines the maximum number of iterative denoising steps allowed for generating the complete sequence. |
| $\eta$ | Noise schedule parameter. This parameter, if applicable, controls the degree of stochasticity or the magnitude of noise introduced during the reverse (denoising) process. Specific values depend on the chosen noise schedule function. |
| $\tau$ | Temperature for sampling. This real-valued parameter influences the randomness of token sampling during generation. Higher values increase randomness, while lower values promote more deterministic (greedy-like) selections. |

# B  EXPERIMENTAL DESIGN

This section details the datasets, experimental parameters, and computational environment employed for evaluating Wavefront Diffusion.

## B.1  DATASETS

Our experiments utilize standard benchmarks for reasoning and code generation tasks.

- **GSM8K**: A dataset for mathematical word problems requiring multi-step reasoning. We use the official test set for evaluation.
- **MATH**: A challenging dataset comprising competition-style math problems across various difficulty levels. We report results on the test set.
- **HumanEval**: A dataset designed for evaluating code generation capabilities, consisting of Python programming problems with test cases. We follow the standard evaluation protocol.
- **Big Bench Hard (BBH)**: A challenging suite of reasoning tasks covering diverse domains such as logic puzzles and multi-step reasoning.

For all datasets, input prompts are formatted according to the specifications of the base Diffusion Language Model.

## B.2 Experimental Parameters

All experiments are conducted using two pre-trained **diffusion-based language models** as the core denoising backbone.

- **Base Model**: LLaDA-8B-Instruct, LLaDA-1.5
- **Sequence Length**: All generated sequences are fixed to a maximum length of 1024 tokens.
- **Number of Diffusion Steps** ($T$): Consistent across all experiments at $T = 1024$ steps.
- **Wavefront Diffusion Parameters**:
  - For all benchmarks, the wavefront size $F$ is set to $8$ and expansion radius $R$ to $2$.
- **Baselines Parameters**: For Standard Diffusion and Block Diffusion baselines, parameters are set to match the respective original implementations or optimized for best performance on these tasks, ensuring a fair comparison. (e.g., 'Block Diffusion block size $B = 8$').

## B.3 Experimental Environment

Experiments were performed on a cluster equipped with 8 NVIDIA A100 GPUs. Each GPU possesses 80 GB of memory. The operating system is Ubuntu 20.04.2 LTS, running Python 3.9.16 with PyTorch 2.7.0. All code was implemented using Hugging Face Transformers library.

## B.4 Example Test Prompts

To enhance reproducibility and clarity, this subsection provides representative test prompts used for each dataset. These examples illustrate the exact input format provided to our Diffusion Language Model.

Table 4: Example test prompts for GSM8K, MATH, and HumanEval datasets. The full set of prompts is included in the released code.

| Dataset | Example Prompt |
| --- | --- |
| GSM8K | <pre>You're an expert at solving elementary math problems involving addition, subtraction, and multiplication... #### x</pre> |
| MATH | <pre>Let's think step by step and output the final answer within boxed{}. Q: {query}</pre> |
| HumanEval | <pre>def has_close_elements(numbers: List[float], threshold: float) -> bool: "Given a list of numbers..." #### True\|False</pre> |

## B.5 Prompt Templates for BBH Subsets

We provide representative prompt templates for subsets of BBH. The templates follow standardized reasoning and answer formats, ensuring reproducibility across tasks.

Table 5: Prompt templates for BBH subsets. Each template enforces explicit reasoning before producing the final answer.

| BBH Subset | Prompt Template |
|---|---|
| `causal_judgement,` `navigate,` `sports_understanding,` `web_of_lies` | `You are deciding \Yes" or \No".` `1) Think step by step...` `####<Yes\|No>` |
| `boolean_expressions` | `You are deciding \True" or \False".` `1) Think step by step...` `####<True\|False>` |
| `formal_fallacies` | `You are deciding \valid" or \invalid".` `1) Think step by step...` `####<valid\|invalid>` |
| `date_understanding,` `disambiguation_qa,` `geometric_shapes,` `hyperbaton,` `logical_deduction_*,` `movie_recommendation,` `penguins_in_a_table,` `reasoning_about_colored_objects,` `ruin_names,` `salient_translation_error_detection,` `snarks,` `temporal_sequences,` `tracking_shuffled_objects_*` | `You are an expert at answering multiple-choice` `questions...` `####(A)\|(B)\|(C)...` |
| `dyck_languages,` `multistep_arithmetic_two,` `object_counting` | `Solve the problem step by step...` `####<Answer>` |
| `word_sorting` | `Solve the problem step by step...` `####<word1 word2 word3 ...>` |

## B.6 Detailed Accuracy of BBH

We report detailed accuracy across BBH subsets for both LLaDA-8B-Instruct and LLaDA-1.5. Base Accuracy refers to standard decoding, Wave Accuracy refers to WavefrontDiffusion decoding, and $\Delta$ denotes Wave – Base.

## B.7 Case Study: Dynamic Wavefront vs. Static Block

To complement the quantitative MHCO analysis in Section 4.4, we present a qualitative case study to visually compare the decoding behaviors of **BlockDiffusion** and **WavefrontDiffusion**. This provides an intuitive understanding of how adaptive scheduling enables better alignment with semantic structure, directly addressing RQ3.

**Setup.** We evaluate both strategies on a code generation task from the `HumanEval` benchmark using the following prompt:

> *"Write a Python function to compute Fibonacci numbers recursively."*

All experiments use the **LLaDA-8B-Instruct** model under a strictly matched compute budget, with deterministic decoding (temperature = 0). No chain-of-thought prompting or auxiliary signals are provided, ensuring that differences arise solely from the decoding schedule.

**Observations.** Table 8 shows snapshots of the decoding process.

Table 6: Accuracy of LLaDA-8B-Instruct across BBH subsets. Δ denotes Wave – Base.

| Task | Base Accuracy | Wave Accuracy | Δ |
|------|---------------|---------------|---|
| causal_judgement | 0.484 | 0.488 | +0.004 |
| temporal_sequences | 0.276 | 0.280 | +0.004 |
| geometric_shapes | 0.176 | 0.288 | +0.112 |
| tracking_shuffled_objects_five_objects | 0.156 | 0.156 | 0.000 |
| salient_translation_error_detection | 0.452 | 0.428 | -0.024 |
| boolean_expressions | 0.832 | 0.844 | +0.012 |
| web_of_lies | 0.544 | 0.548 | +0.004 |
| formal_fallacies | 0.548 | 0.576 | +0.028 |
| tracking_shuffled_objects_seven_objects | 0.112 | 0.128 | +0.016 |
| tracking_shuffled_objects_three_objects | 0.296 | 0.328 | +0.032 |
| ruin_names | 0.400 | 0.420 | +0.020 |
| dyck_languages | 0.012 | 0.008 | -0.004 |
| movie_recommendation | 0.440 | 0.408 | -0.032 |
| snarks | 0.412 | 0.356 | -0.056 |
| navigate | 0.684 | 0.672 | -0.012 |
| disambiguation_qa | 0.408 | 0.420 | +0.012 |
| hyperbaton | 0.516 | 0.500 | -0.016 |
| object_counting | 0.704 | 0.660 | -0.044 |
| reasoning_about_colored_objects | 0.452 | 0.424 | -0.028 |
| logical_deduction_five_objects | 0.348 | 0.380 | +0.032 |
| date_understanding | 0.424 | 0.412 | -0.012 |
| penguins_in_a_table | 0.300 | 0.344 | +0.044 |
| logical_deduction_seven_objects | 0.436 | 0.448 | +0.012 |
| multistep_arithmetic_two | 0.476 | 0.568 | +0.092 |
| logical_deduction_three_objects | 0.500 | 0.556 | +0.056 |

Table 7: Accuracy of LLaDA-1.5 across BBH subsets. Δ denotes Wave – Base.

| Task | Base Accuracy | Wave Accuracy | Δ |
|------|---------------|---------------|---|
| movie_recommendation | 0.416 | 0.392 | -0.024 |
| reasoning_about_colored_objects | 0.496 | 0.508 | +0.012 |
| disambiguation_qa | 0.428 | 0.428 | 0.000 |
| web_of_lies | 0.556 | 0.524 | -0.032 |
| formal_fallacies | 0.556 | 0.568 | +0.012 |
| date_understanding | 0.420 | 0.380 | -0.040 |
| ruin_names | 0.364 | 0.444 | +0.080 |
| temporal_sequences | 0.280 | 0.288 | +0.008 |
| geometric_shapes | 0.252 | 0.356 | +0.104 |
| tracking_shuffled_objects_seven_objects | 0.124 | 0.124 | 0.000 |
| logical_deduction_seven_objects | 0.476 | 0.468 | -0.008 |
| dyck_languages | 0.024 | 0.028 | +0.004 |
| salient_translation_error_detection | 0.460 | 0.496 | +0.036 |
| logical_deduction_five_objects | 0.408 | 0.364 | -0.044 |
| logical_deduction_three_objects | 0.516 | 0.596 | +0.080 |
| snarks | 0.368 | 0.384 | +0.016 |
| penguins_in_a_table | 0.348 | 0.348 | 0.000 |
| tracking_shuffled_objects_three_objects | 0.344 | 0.396 | +0.052 |
| object_counting | 0.684 | 0.668 | -0.016 |
| multistep_arithmetic_two | 0.424 | 0.484 | +0.060 |
| tracking_shuffled_objects_five_objects | 0.180 | 0.124 | -0.056 |
| boolean_expressions | 0.852 | 0.836 | -0.016 |
| hyperbaton | 0.512 | 0.496 | -0.016 |
| navigate | 0.724 | 0.676 | -0.048 |
| causal_judgement | 0.484 | 0.488 | +0.004 |

Table 8: **Case Study:** Decoding snapshots for the same task at matched compute. Column 1: decoding step $t$. Column 2: **BlockDiffusion** outputs, progressing strictly left-to-right. Column 3: **WavefrontDiffusion** outputs, dynamically expanding to semantically relevant regions.

| Step | BlockDiffusion | WavefrontDiffusion |
|------|----------------|--------------------|
| $t = 1$ | `def fib(` | `def fib(` **+ partial** `if` clause |
| $t = 2$ | `def fib(n):  if` | `def fib(n):  if n <= 1:` **+ partial** `return` |
| $t = 3$ | `def fib(n):  if n <= 1:` | `def fib(n):  if n <= 1:  return n` |
| $t = 4$ | `return fib(n-1) + fib(n-2)` | `return fib(n-1) + fib(n-2)` |

At the initial step ($t = 1$), **BlockDiffusion** begins strictly from the left, decoding only the initial function signature. In contrast, **WavefrontDiffusion** simultaneously expands outward, partially decoding both the signature and the beginning of the `if` clause.

By $t = 2$, BlockDiffusion still has not completed the `if` condition, resulting in an incomplete logical structure. WavefrontDiffusion, however, has already produced a nearly complete condition and begun generating the return statement.

At $t = 3$, WavefrontDiffusion fully resolves the core recursive structure (`return n` or recursive calls), while BlockDiffusion has just finished the condition.

Finally, at $t = 4$, both methods converge to the same final implementation, but WavefrontDiffusion reached a semantically coherent intermediate state much earlier.

**Implication.** This case study demonstrates how WavefrontDiffusion avoids the artificial barriers imposed by fixed blocks. By flexibly expanding to semantically related regions, it maintains coherent partial outputs throughout the generation process. This visual evidence complements our MHCO analysis and explains the accuracy improvements observed in Table 1, highlighting how adaptive scheduling aligns decoding with the natural structure of reasoning and code.

## C  ADDITIONAL EXPERIMENTAL RESULTS

### C.1  COMPARISON WITH ADVANCED DYNAMIC BASELINES

To further validate the effectiveness of WavefrontDiffusion, we extend our comparison to include three recent advanced decoding strategies that also incorporate dynamic or adaptive mechanisms. These methods represent the state-of-the-art in optimizing diffusion decoding beyond fixed schedules:

- **Truncated-BlockDiffusion (TBD)** (Anonymous, 2025): A concurrent approach that adaptively adjusts the block size during generation based on local confidence, aiming to solve the rigidity of fixed blocks.

- **Running Confidence Remasking (RCR)** (He et al., 2025): A training-free strategy proposed in MDPO that identifies and re-masks low-confidence tokens during the decoding process, allowing the model to revise potential errors.

- **Self-Speculative Decoding (SSD)** (Gao et al., 2025): A method designed primarily for acceleration via hierarchical verification, which effectively acts as a dynamic acceptance policy for generated tokens.

**Results.** We conduct the comparison using **LLaDA-8B-Instruct** with the total step budget fixed at $T = 1024$. As shown in Table 9, WavefrontDiffusion consistently outperforms these advanced baselines across all four benchmarks. While TBD and RCR improve upon Standard Diffusion, they still lag behind WavefrontDiffusion in complex reasoning tasks (e.g., MATH and GSM8K). This suggests that while adaptive block sizing (TBD) and remasking (RCR) are beneficial, the *wavefront* expansion strategy—which aligns update order with semantic dependencies—provides a more optimal decoding path for reasoning-heavy generation.

Table 9: **Comparison with Advanced Dynamic Baselines.** Accuracy on LLaDA-8B-Instruct ($T = 1024$). WavefrontDiffusion achieves state-of-the-art performance among training-free dynamic strategies.

| Method | GSM8K | MATH | HumanEval | BBH |
|---|---|---|---|---|
| Standard Diffusion | 23.15 | 26.60 | 17.68 | 11.30 |
| BlockDiffusion | 80.74 | 40.62 | 45.73 | 43.23 |
| Truncated-BlockDiffusion (TBD) | 80.90 | 40.84 | 45.73 | 39.41 |
| Running Confidence Remasking (RCR) | 80.60 | 40.02 | 43.90 | 43.10 |
| Self-Speculative Decoding (SSD) | 79.15 | 38.88 | 44.51 | 41.25 |
| **WavefrontDiffusion (Ours)** | **82.03** | **41.04** | **47.56** | **44.30** |

## C.2 EFFICIENCY AND COMPUTATIONAL PARITY ANALYSIS

A key concern in iterative decoding is whether performance gains come at the cost of increased latency. We clarify that WavefrontDiffusion is a *scheduling strategy*, not an acceleration method, designed to align strictly with the computational budget of standard block-based decoding.

**Computational Parity.** As detailed in Table 10, our method shares the identical number of denoising steps ($T = 1024$) and model configuration as BlockDiffusion. WavefrontDiffusion alters the *order* of token updates but not the *quantity* of forward passes. Consequently, the total FLOPs per sample remain equivalent ($2.517 \times 10^{15}$ for LLaDA-8B). Observed wall-clock time differences are within $\pm 2\%$, attributed solely to system variance rather than algorithmic overhead.

Table 10: **Compute-Parity Comparison.** We report FLOPs, total inference time (s), and throughput (Tokens Per Second, TPS) on LLaDA-8B and LLaDA-1.5 ($T = 1024$). WavefrontDiffusion maintains identical computational cost to BlockDiffusion while achieving higher accuracy.

| Model | Strategy | FLOPs ($\times 10^{15}$) | GSM8K Time | GSM8K TPS | MATH Time | MATH TPS | HumanEval Time | HumanEval TPS | BBH Time | BBH TPS |
|---|---|---|---|---|---|---|---|---|---|---|
| **LLaDA-8B** | Standard Diffusion | 2.517 | 110.1 | 9.30 | 108.8 | 9.41 | 115.8 | 8.84 | 122.7 | 8.35 |
| | BlockDiffusion | 2.517 | 113.6 | 9.02 | 109.4 | 9.36 | 107.2 | 9.55 | 116.6 | 8.78 |
| | **WavefrontDiffusion** | 2.517 | 112.2 | 9.13 | 109.9 | 9.32 | 109.3 | 9.37 | 115.1 | 8.89 |
| **LLaDA-1.5** | Standard Diffusion | 2.517 | 111.9 | 9.15 | 111.4 | 9.19 | 118.0 | 8.68 | 127.2 | 8.05 |
| | BlockDiffusion | 2.517 | 111.6 | 9.17 | 107.4 | 9.53 | 110.1 | 9.30 | 118.6 | 8.63 |
| | **WavefrontDiffusion** | 2.517 | 109.3 | 9.37 | 110.9 | 9.23 | 112.9 | 9.07 | 114.1 | 8.97 |

**Compatibility with Acceleration.** Although WavefrontDiffusion itself does not aim to accelerate inference, it is fully compatible with existing acceleration frameworks. Specifically:

- **KV-Cache Optimization**: Since our method only changes the masking schedule, it integrates seamlessly with memory-reuse techniques like Fast-dLLM (Wu et al., 2025).

- **Speculative Decoding**: The adaptive wavefront expansion is orthogonal to sampling compression and hierarchical verification methods (Gao et al., 2025), allowing for combined use to achieve both quality improvements and latency reduction.

## C.3 ABLATION STUDY ON DENOISING STEPS

To investigate how the number of denoising steps affects generation quality, we performed a controlled ablation on **LLaDA-8B-Instruct** using the **GSM8K** benchmark. We varied the total diffusion steps $T \in \{256, 512, 1024\}$, which correspond to finalize quotas $k_t \in \{4, 2, 1\}$ respectively.

As shown in Table 11, WavefrontDiffusion remains competitive even at lower denoising steps and consistently surpasses BlockDiffusion as $T$ increases. The relative improvement becomes more

pronounced at higher $T$, indicating that the adaptive wavefront scheduling effectively leverages extended denoising horizons to propagate semantic dependencies.

Table 11: **Ablation on Denoising Steps** ($T$). Accuracy on GSM8K using LLaDA-8B-Instruct. WavefrontDiffusion scales effectively with denoising depth.

| Strategy | Steps = 256 | Steps = 512 | Steps = 1024 |
|---|---|---|---|
| Standard Diffusion | 12.96 | 17.05 | 23.15 |
| BlockDiffusion | **43.14** | 67.02 | 80.74 |
| **WavefrontDiffusion (Ours)** | 42.38 | **73.79** | **82.03** |

## C.4 ROBUSTNESS ANALYSIS: TEMPERATURE AND RANDOM SEEDS

We further evaluated the robustness of WavefrontDiffusion under stochastic sampling conditions. We conducted experiments on **GSM8K** (LLaDA-8B), varying the sampling temperature from 0.0 to 0.8. Each value represents the mean accuracy $\pm$ standard deviation over three random seeds.

Table 12 demonstrates that WavefrontDiffusion maintains superior performance and low variance across all temperature regimes. The consistent advantage over BlockDiffusion (approx. +1.6%– 2.0%) indicates that our dynamic frontier scheduling is robust to sampling randomness.

Table 12: **Temperature Robustness Analysis.** Mean accuracy $\pm$ standard deviation on GSM8K (LLaDA-8B).

| Strategy | Temp = 0.0 | Temp = 0.3 | Temp = 0.6 | Temp = 0.8 |
|---|---|---|---|---|
| Standard Diffusion | $23.15 \pm 0.00$ | $23.42 \pm 0.15$ | $22.97 \pm 0.21$ | $20.35 \pm 0.22$ |
| BlockDiffusion | $80.74 \pm 0.00$ | $80.83 \pm 0.14$ | $81.09 \pm 0.20$ | $78.42 \pm 0.37$ |
| **WavefrontDiffusion** | $\mathbf{82.03 \pm 0.00}$ | $\mathbf{81.47 \pm 0.12}$ | $\mathbf{81.97 \pm 0.26}$ | $\mathbf{78.86 \pm 0.41}$ |

## C.5 EFFECTIVENESS WITH ALTERNATIVE TOKEN-PRIORITY METRICS

To verify that our method's success is not dependent solely on confidence scores, we evaluated WavefrontDiffusion using two alternative frontier-priority metrics on the **Dream-7B** model:

1. **Entropy**: Lower entropy indicates higher priority.

2. **Margin**: Larger gap between top-1 and top-2 probabilities indicates higher priority.

As shown in Table 13, WavefrontDiffusion consistently outperforms baselines regardless of the priority metric used, confirming the universality of the wavefront scheduling mechanism.

Table 13: **Performance under Alternative Token-Priority Metrics.** Experiments on Dream-7B with equal FLOPs ($T = 1024$).

| Strategy | Metric | GSM8K | MATH | HumanEval | BBH |
|---|---|---|---|---|---|
| Standard Diffusion | Entropy | 34.80 | 29.56 | 20.13 | 15.90 |
| BlockDiffusion | Entropy | 81.96 | 42.26 | 50.61 | 44.63 |
| **WavefrontDiffusion** | Entropy | **82.79** | **42.90** | **52.43** | **44.71** |
| Standard Diffusion | Margin | 34.90 | 29.28 | 20.73 | 16.10 |
| BlockDiffusion | Margin | 78.85 | 41.16 | 50.00 | 44.90 |
| **WavefrontDiffusion** | Margin | **79.38** | **42.08** | **50.61** | **45.35** |

## C.6 Matched-Latency Comparison with Autoregressive Models

Finally, we provide a latency-matched comparison against Autoregressive (AR) greedy decoding and Self-Speculative Decoding (SSD). Table 14 shows that WavefrontDiffusion achieves comparable latency to AR decoding while delivering superior accuracy, demonstrating its practical viability.

Table 14: **Matched-Latency Comparison.** LLaDA-8B-Instruct on GSM8K.

| Method | Latency (ms/token) ↓ | Accuracy ↑ |
|---|---|---|
| Autoregressive Greedy | 9.2 | 80.67 |
| Self-Speculative Decoding | 5.8 | 79.30 |
| **WavefrontDiffusion (Ours)** | **9.1** | **82.03** |

## C.7 Calibration Analysis

Since WavefrontDiffusion relies on confidence scores to determine the decoding frontier, validating that these scores are well-calibrated is critical. We conducted a rigorous calibration analysis to ensure that the model's internal confidence serves as a reliable signal for decision-making.

**Experimental Setup.** To measure intrinsic calibration without the confounding factor of divergent reasoning paths, we employed a standard **Teacher Forcing** protocol. We fed the model the ground truth prefix at each step and recorded the confidence of tokens selected by WavefrontDiffusion versus BlockDiffusion. We report the **Expected Calibration Error (ECE)** and **Maximum Calibration Error (MCE)** on GSM8K, MATH, and HumanEval.

**Quantitative Results.** Table 15 summarizes the calibration metrics. WavefrontDiffusion achieves consistently lower ECE scores across all domains, with a reduction of approximately **40%** on GSM8K ($0.1093 \rightarrow 0.0659$). Furthermore, it significantly reduces the Maximum Calibration Error (MCE) on complex tasks like MATH and HumanEval ($0.7051 \rightarrow 0.4502$). This indicates that our adaptive wavefront effectively filters out poorly calibrated tokens that static block schedules are forced to commit to.

Table 15: **Calibration Metrics.** Comparison of Expected Calibration Error (ECE) and Maximum Calibration Error (MCE) under Teacher Forcing. Lower is better.

| Dataset | Method | ECE (↓) | MCE (↓) |
|---|---|---|---|
| **GSM8K** | BlockDiffusion | 0.1093 | **0.4727** |
| | **WavefrontDiffusion (Ours)** | **0.0659** | 0.4824 |
| **MATH** | BlockDiffusion | 0.1025 | 0.3457 |
| | **WavefrontDiffusion (Ours)** | **0.0901** | **0.2393** |
| **HumanEval** | BlockDiffusion | 0.0920 | 0.7051 |
| | **WavefrontDiffusion (Ours)** | **0.0853** | **0.4502** |

**Reliability Diagrams.** Figure 3 visualizes the reliability diagrams for both methods across the three datasets. Ideally, the calibration curve should align with the diagonal (dashed line). Comparing the left column (BlockDiffusion) and the right column (WavefrontDiffusion), we observe that WavefrontDiffusion's curves are consistently closer to the diagonal, particularly in high-confidence regions. This confirms that the tokens selected by the wavefront are fundamentally more "trustworthy," validating the reliability of our adaptive scheduling mechanism.

## C.8 Decoding Dynamics Analysis

To explicitly address the concern regarding frontier evolution and stability, we visualize the step-wise decoding dynamics in Figure 4. This analysis captures the "process" of generation, contrasting the rigid scheduling of BlockDiffusion with the adaptive flow of WavefrontDiffusion.

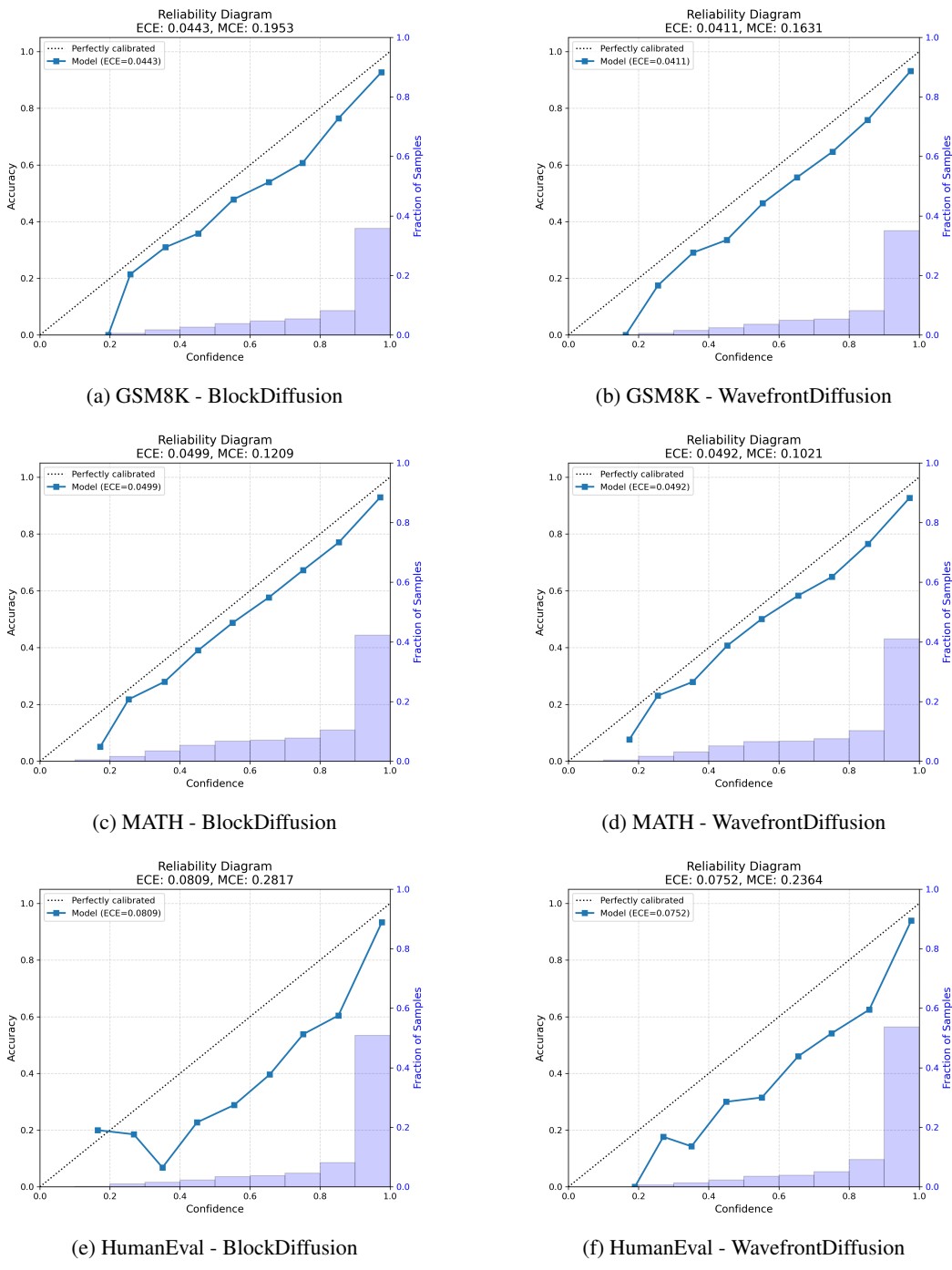

Figure 3: **Reliability Diagrams Comparison.** Calibration curves for BlockDiffusion (Left) and WavefrontDiffusion (Right) across GSM8K, MATH, and HumanEval. The dashed diagonal line represents perfect calibration. WavefrontDiffusion exhibits curves that are consistently tighter to the diagonal, indicating superior calibration.

**1. Frontier Evolution (Smoothness vs. Stagnation).** As shown in Figure 4a, BlockDiffusion exhibits a characteristic "staircase" pattern, where the frontier freezes within a block until all tokens are finalized, regardless of their difficulty. In contrast, WavefrontDiffusion (green curve) demonstrates a *smooth, continuous progression*. The absence of horizontal plateaus confirms that our method avoids premature freezing and adapts the generation pace naturally to the sequence structure.

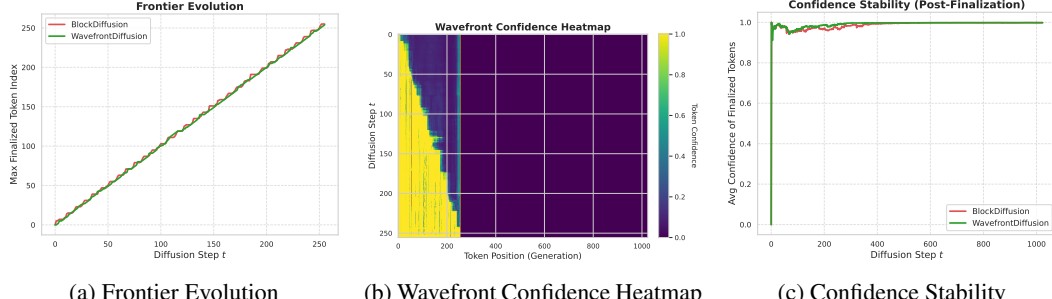

| (a) Frontier Evolution | (b) Wavefront Confidence Heatmap | (c) Confidence Stability |

Figure 4: **Decoding Dynamics Analysis.** (a) WavefrontDiffusion shows a smooth, continuous progression (green), avoiding the rigid "staircase" artifacts of BlockDiffusion (red). (b) The heatmap reveals a clear "diagonal frontier" where the wavefront adapts its velocity to token difficulty. (c) WavefrontDiffusion maintains consistently high post-finalization confidence, indicating minimal "regret," whereas BlockDiffusion suffers from confidence drops due to insufficient lookahead context.

**2. The "Wave" Mechanism (Adaptivity).** Figure 4b visualizes the confidence heatmap of WavefrontDiffusion. The bright diagonal band represents the active wavefront. Notably, the boundary is not a perfect straight line but exhibits local fluctuations. This visualizes the *adaptivity* of the algorithm: the wavefront accelerates through high-confidence (easy) regions and slows down to accumulate context for low-confidence (hard) tokens, effectively implementing a dynamic computation allocation.

**3. Post-Finalization Stability (Regret Analysis).** In Figure 4c, we track the average confidence of tokens *after* they are finalized. Ideally, a model should not "regret" its decisions (i.e., confidence should not drop after finalizing). BlockDiffusion (red) shows jitter and drops, implying that forced block commits often lead to sub-optimal decisions that the model later finds inconsistent. WavefrontDiffusion (green) maintains a stable, near-1.0 confidence trajectory, proving that its context-aware expansion significantly reduces finalization regret.

## D    THEORETICAL JUSTIFICATION

In this section, we provide the formal theoretical framework that justifies WavefrontDiffusion as the optimal expansion of restricted-search decoding strategies. Our derivation rests on two core pillars: the Information Gradient Hypothesis and the optimality of dynamic restricted search.

### D.1    PREMISE 1: THE INFORMATION GRADIENT HYPOTHESIS

Let $\mathcal{C}_t$ be the set of finalized tokens at step $t$. For any masked token $i$, let $d(i, \mathcal{C}_t)$ be the minimum distance to the finalized context (as defined in Eq. 1). We introduce the **Information Gradient Assumption**, which states that the conditional entropy $H$ of a masked token $x_i$ is monotonically non-decreasing with respect to its distance from the finalized context:

$$H(x_i|x_{\mathcal{C}_t}) \leq H(x_j|x_{\mathcal{C}_t}) \iff d(i, \mathcal{C}_t) < d(j, \mathcal{C}_t) \tag{5}$$

This formalizes the intuition that tokens strictly adjacent to the known context possess the highest "information density" and thus the highest probability of being correctly predicted.

### D.2    PREMISE 2: THE VALIDITY OF RESTRICTED SEARCH

The global optimization objective at step $t$ is to select a subset of tokens $S_t$ (where $|S_t| = K$) to finalize such that we maximize joint confidence. BlockDiffusion (Arriola et al., 2025) demonstrated a key theoretical contribution: **Global search is unnecessary**. It proved that limiting the search space to a constrained subset $\Omega \subset \mathcal{V}_{all}$ yields efficient results, provided the subset contains valid candidates.

- **BlockDiffusion's Constraint:** $\Omega_{Block} = \{i \mid i \in \text{Block}_k\}$.
- **Limitation:** Because $\Omega_{Block}$ is static, there exists a non-zero probability of "semantic mismatch," where true high-confidence tokens lie outside the block. That is, $\exists j \notin \Omega_{Block}$ such that $H(x_j) < \min_{i \in \Omega_{Block}} H(x_i)$.

### D.3 THEOREM: WAVEFRONTDIFFUSION AS OPTIMAL RESTRICTED SEARCH

Our method aligns the restriction boundary $\Omega$ with the Information Gradient defined in Premise 1. We define the Wavefront search space $\Omega_{Wave}$ as:

$$\Omega_{Wave} = \{i \mid d(i, \mathcal{C}_t) \leq R\} \tag{6}$$

**Theorem.** Under the Information Gradient Assumption, the optimal subset $S^*$ (the set of $K$ tokens with lowest entropy) satisfies $S^* \subseteq \Omega_{Wave}$ for a sufficient radius $R$, whereas $S^* \subseteq \Omega_{Block}$ is not guaranteed.

**Proof Sketch.** Since entropy $H(x_i)$ increases with distance $d(i, \mathcal{C}_t)$, the "lowest entropy isosurfaces" form expanding contours around $\mathcal{C}_t$:

1. BlockDiffusion forces a search within a rectangular window $\text{Block}_k$. This window inevitably intersects with high-entropy regions (far from context) while potentially excluding low-entropy regions (near context but in the next block) due to rigid segmentation.

2. WavefrontDiffusion strictly defines $\Omega_{Wave}$ to encapsulate the lowest distance $d$ values. Therefore, $\Omega_{Wave}$ is mathematically guaranteed to contain the highest density of correctable tokens.

3. Consequently, WavefrontDiffusion ensures that the local optimization is performed within the theoretical upper bound of candidate quality, extending the efficiency logic of BlockDiffusion to its optimal semantic conclusion.

**Empirical Verification.** This theoretical proof is empirically supported by our MHCO analysis in Section 4.4. The significantly lower MHCO scores for WavefrontDiffusion confirm that our dynamic boundary successfully captures the optimal local search space defined by this framework, whereas BlockDiffusion's static boundary frequently violates it.

